# Towards an Image-Informed Mathematical Model of In Vivo Response to Fractionated Radiation Therapy

**DOI:** 10.3390/cancers13081765

**Published:** 2021-04-07

**Authors:** David A. Hormuth, Angela M. Jarrett, Tessa Davis, Thomas E. Yankeelov

**Affiliations:** 1Oden Institute for Computational Engineering and Sciences, The University of Texas at Austin, Austin, TX 78712, USA; ajarrett@utexas.edu (A.M.J.); thomas.yankeelov@utexas.edu (T.E.Y.); 2Livestrong Cancer Institutes, The University of Texas at Austin, Austin, TX 78712, USA; 3Department of Biomedical Engineering, The University of Texas at Austin, Austin, TX 78712, USA; tessa@austin.utexas.edu; 4Department of Diagnostic Medicine, The University of Texas at Austin, Austin, TX 78712, USA; 5Department of Oncology, The University of Texas at Austin, Austin, TX 78712, USA; 6Department of Imaging Physics, The University of Texas MD Anderson Cancer Center, Houston, TX 77030, USA

**Keywords:** computational oncology, magnetic resonance imaging, perfusion, glioblastoma, U87

## Abstract

**Simple Summary:**

Using medical imaging data and computational models, we develop a modeling framework to provide personalized treatment response forecasts to fractionated radiation therapy for individual tumors. We evaluate this approach in an animal model of brain cancer and forecast changes in tumor cellularity and vasculature.

**Abstract:**

Fractionated radiation therapy is central to the treatment of numerous malignancies, including high-grade gliomas where complete surgical resection is often impractical due to its highly invasive nature. Development of approaches to forecast response to fractionated radiation therapy may provide the ability to optimize or adapt treatment plans for radiotherapy. Towards this end, we have developed a family of 18 biologically-based mathematical models describing the response of both tumor and vasculature to fractionated radiation therapy. Importantly, these models can be personalized for individual tumors via quantitative imaging measurements. To evaluate this family of models, rats (n = 7) with U-87 glioblastomas were imaged with magnetic resonance imaging (MRI) before, during, and after treatment with fractionated radiotherapy (with doses of either 2 Gy/day or 4 Gy/day for up to 10 days). Estimates of tumor and blood volume fractions, provided by diffusion-weighted MRI and dynamic contrast-enhanced MRI, respectively, were used to calibrate tumor-specific model parameters. The Akaike Information Criterion was employed to select the most parsimonious model and determine an ensemble averaged model, and the resulting forecasts were evaluated at the global and local level. At the global level, the selected model’s forecast resulted in less than 16.2% error in tumor volume estimates. At the local (voxel) level, the median Pearson correlation coefficient across all prediction time points ranged from 0.57 to 0.87 for all animals. While the ensemble average forecast resulted in increased error (ranging from 4.0% to 1063%) in tumor volume predictions over the selected model, it increased the voxel wise correlation (by greater than 12.3%) for three of the animals. This study demonstrates the feasibility of calibrating a model of response by serial quantitative MRI data collected during fractionated radiotherapy to predict response at the conclusion of treatment.

## 1. Introduction

With over a century of development, radiotherapy (RT) remains a principal component of oncological care for many disease sites including brain cancer. In the particular case of high-grade gliomas, RT is administered alongside chemotherapy to treat residual and unresected disease [1]. The delivery of highly conformal RT doses has been enabled by the development of intensity-modulated RT and, more recently, magnetic resonance imaging (MRI) guided linear accelerators [2]. These conformal RT techniques can deliver the prescribed dose to the tumor while significantly reducing the dose delivered to healthy appearing tissue [3]. However, the full potential of RT techniques have not yet been fully realized as current RT treatment plans are based predominately on anatomical or structural imaging measures that report on the extent of the disease, rather than the underlying biology which may provide more fundamental information on treatment response [4]. Several quantitative imaging techniques have emerged as potential imaging biomarkers [4] to adapt or boost RT plans [5]; however, they still only provide a temporal snapshot of the tumor biology. Imaged-based mathematical modeling techniques [6] may provide a means to forecast tumor dynamics in response to RT, and thereby enable the personalization of RT plans for each patient’s unique tumor biology. 

Cell survival following RT is commonly described by the linear quadratic (LQ) model, which reports the fraction of cells that will survive and proliferate to form a colony of cells given a specific radiation dose and cell-specific radiosensitivity parameters [7]. One strength of the LQ model is its simplicity for evaluating the efficacy of different RT regimens for tumor and healthy appearing tissue, thus enabling the selection of fractionated RT doses that have the most therapeutic benefit. One shortcoming, however, is that it does not capture the temporal dynamics of tumor response. That is, following a single fraction of RT the LQ model cannot estimate when a tumor will shrink or when tumor cells will die. Response to RT is not binary (i.e., cells do not either die or emerge unscathed from exposure to radiation), rather there is a spectrum of response ranging from cells that receive sufficient damage to undergo apoptosis, to cells that continue proliferating for a few generations before becoming senescent, or cells that proliferate indefinitely due to successful DNA repair [8]. In recent years, we [9] and others [10,11,12,13] have attempted to connect models capturing the dynamics of tumor growth with the efficacy of RT. Beyond characterizing response to RT, these mechanism-based and machine learning techniques can also be used to predict response [9,14], identify optimal treatment strategies [15,16,17,18], and evaluate alternative schedules based off different growth assumptions [19]. In the works of Rockne et al. [13] and Prokopiou et al. [12], the effects of RT were assumed to result in the immediate loss of tumor cells (or reduction in tumor volume) as estimated from the LQ model. In a leave-one-out cross validation, Rockne et al. observed that the predicted post-RT tumor volume was well within the inter-observer tumor volume uncertainty [13] demonstrating a potential approach to evaluating the efficacy of RT regimens. Prokopiou et al. [12] proposed the use of a proliferation saturation index to identify patients that might benefit from non-standard fractionation regimens; this modeling framework was later shown to provide strong predictive accuracy of future response [20]. Alternatively, non-instantaneous cell death following RT was considered by Brüningk et al. [11] and Hormuth et al. [9] in their models of response in the in vitro and in vivo pre-clinical setting, respectively. Both Brüningk et al. [11] and Hormuth et al. [9] observed that models with an instantaneous cell death greatly increased the error in model fits and/or predictions compared to models with a delayed death or altered proliferation, respectively. However, it should be noted that models of instantaneous death may be more appropriate for longer time scales (i.e., days to weeks) when sufficient time has passed for the tumor to shrink. While these efforts have shown promising results capturing the temporal dynamics of response to RT, techniques capable of also predicting the spatial response to fractionated RT are still lacking. Towards this goal, we have extended our image-informed modeling approach [6,9,21] to consider the response of tumor and vasculature to fractionated RT in a pre-clinical model of high-grade glioma.

Our previous modeling studies considered single, large dose, RT [9], which does not mirror the standard-of-care for high-grade glioma patients which consists of 2 Gy per weekday for 6 weeks [22]. To address this limitation, we revised our experimental approach to delivery 2 Gy or 4 Gy per weekday for up to 2 weeks. Additionally, animals were imaged with quantitative magnetic resonance imaging (MRI) techniques sensitive to tumor cellularity and tumor vasculature [23,24] before, during, and following the completion of RT. By using quantitative MRI, we are able to non-invasively observe and quantify tumor physiology in 3D throughout the study, which is then used for calibration of animal specific tumor growth and response parameters as well as validation of predicted tumor response to treatment. From the observed dynamics of tumor growth and response, we developed three models of response to RT. The first model assumed immediate cell death and removal (similar to the approaches used by [12,13]). The second model assumed immediate cell death and a reduction of tumor proliferation rate (due to cell cycle arrest), while the third model assumed a reduction of tumor proliferation rate and an increase in tumor cell death rate (due to delayed mitotic cell death). The three models of response to RT were incorporated in a family of 18 models that included three approaches to spatially-vary the efficacy of RT and two approaches to parameterizing the tumor cell proliferation rate. In this preliminary study, we calibrated the entire family of 18 models to each individual animal and evaluated the descriptive (i.e., how well the model fits the data) and predictive (i.e., how well the model predicts future response) accuracy of these models. We then used the Akaike Information Criterion (AIC) to select the best or most parsimonious model and calculate model weights. We then quantified the descriptive and predictive error for both the selected model and the ensemble averaged model. 

## 2. Materials and Methods

### 2.1. Theory

#### 2.1.1. Mathematical Description of Tumor and Vasculature Growth

Tumor and vasculature growth are described using a coupled set of 3D partial differential equations built upon the reaction-diffusion model that has been extensively studied at the pre-clinical and clinical settings in glioma [9,25,26,27] and other tumors [28,29]. In our previous efforts, we have applied this model in the setting of untreated tumor growth [21] and single fraction radiotherapy [9]. Here, it is used as the base model upon which we build a set of models of response to fractionated radiotherapy (described in the Section 2.2). We now present the salient features of the model which captures tumor and vasculature growth in the absence of treatment and list the model parameters and variables in Table 1. (The extension of the model to account for the effects of radiation are described below in Section 2.1.2) While a more detailed description of the development and model assumptions are found elsewhere [21], Equation (1) is the main operational equation and describes the spatial and temporal evolution of the tumor volume fraction: (1)$$\begin{array}{ll} \frac{\partial \phi_{T}\left(\bar{x},t\right)}{\partial t}= & \overset{Diffusion}{\overset{︷}{\nabla \cdot \left(D_{T}\left(\bar{x},t\right)\cdot \left(\left(\left(1-\frac{\phi_{V}\left(\bar{x},t\right)}{\theta_{T,V}\left(\bar{x},t\right)}\right)\nabla \frac{\phi_{T}\left(\bar{x},t\right)}{\theta_{T,V}\left(\bar{x},t\right)}\right)+\left(\frac{\phi_{T}\left(\bar{x},t\right)}{\theta_{T,V}\left(\bar{x},t\right)}\nabla \frac{\phi_{V}\left(\bar{x},t\right)}{\theta_{T,V}\left(\bar{x},t\right)}\right)\right)\right)}}\\ & +\overset{Logistic\;Growth}{\overset{︷}{k_{p,T}\phi_{T}\left(\bar{x},t\right)\left(1-\phi_{T}\left(\bar{x},t\right)/\theta_{T}\left(\bar{x},t\right)\right)}} \end{array}$$
where $\phi_{T}\left(\bar{x},t\right)$ is the tumor cell volume fraction at three-dimensional position $\bar{x}$ and time *t*, $D_{T}\left(\bar{x},t\right)$ is the tumor cell diffusion coefficient coupled to local mechanical properties, $\phi_{V}\left(\bar{x},t\right)$ is the blood volume fraction, $\theta_{T,V}\left(\bar{x},t\right)$ is the summation of tumor and the blood volume fraction carrying capacities, *k_p,T_* is the tumor cell proliferation rate, and $\theta_{T}\left(\bar{x},t\right)$ is the tumor cell carrying capacity (i.e., the maximum packing fraction that a voxel can functionally support). We assume tumor cell diffusion (the first term on the right hand side of Equation (1)) is influenced both by the space occupied by tumor associated vasculature (i.e., $\phi_{V}$ terms) as well as local mechanical stress that alter $D_{T}\left(\bar{x},t\right)$ according to: (2)$$D_{T}(\bar{x},t)=D_{T,0}\exp\left(-\lambda_{1}\cdot \sigma_{vm}(\bar{x},t)\right)$$

In Equation (2), *D_T,_*_0_ is the unrestricted (or maximal) tumor cell diffusion coefficient in the absence of stress, $\lambda_{1}$ is the stress-tumor cell diffusion coupling constant, and $\sigma_{vm}(\bar{x},t)$ is the von Mises stress. The von Mises stress is used as it reflects the total stress experienced for a given section of tissue, and is determined by solving for tissue displacement, $\vec{u}$, using the linear elastic, isotropic equilibrium equation defined as:(3)$$\nabla \cdot G\nabla \vec{u}+\nabla \frac{G}{1-2v}\left(\nabla \cdot \vec{u}\right)-\lambda_{2}\nabla \phi_{T}(\bar{x},t)=0$$
where *G* is the shear modulus, *ν* is the Poisson’s ratio, and $\lambda_{2}$ is the second coupling constant. Literature values [30] are used to assign tissue specific *G* and *ν*.

Our previous modeling study [21] identified that the optimal coupling of tumor vasculature to tumor cell growth was via the carrying capacity $\theta_{T}\left(\bar{x},t\right)$. Thus, $\theta_{T}\left(\bar{x},t\right)$ is evolved spatially and temporally in response to changes in the local tumor vasculature according to: (4)$$\theta_{T}\left(\bar{x},t\right)=\{\begin{array}{cc} \theta_{\max} & \phi_{V}\left(\bar{x},t\right)\geq \phi_{V,thresh}\\ \theta_{\min}+\phi_{V}\left(\bar{x},t\right)\left(\frac{\theta_{\max}-\theta_{\min}}{\phi_{V,thresh}}\right) & \phi_{V}\left(\bar{x},t\right)<\phi_{V,thresh} \end{array}$$
where the range of expected carrying capacities is from $\theta_{\min}$ to $\theta_{\max}$, and $\phi_{V,thresh}$ represents a critical value for tumor vasculature that would begin to change the number of cells a voxel can support. The parameters $\theta_{\min}$ and $\theta_{\max}$ are assigned as the lowest and highest volume fractions, respectively, observed during the image visits used for calibration. In a similar fashion, we describe the spatial-temporal evolution of tumor vasculature using: (5)$$\begin{array}{ll} \frac{\partial \phi_{V}\left(\bar{x},t\right)}{\partial t}= & \overset{Diffusion}{\overset{︷}{\nabla \cdot \left(D_{V}\left(\bar{x},t\right)\cdot \left(\left(\left(1-\frac{\phi_{T}\left(\bar{x},t\right)}{\theta_{T,V}\left(\bar{x},t\right)}\right)\nabla \frac{\phi_{V}\left(\bar{x},t\right)}{\theta_{T,V}\left(\bar{x},t\right)}\right)+\left(\frac{\phi_{V}\left(\bar{x},t\right)}{\theta_{T,V}\left(\bar{x},t\right)}\nabla \frac{\phi_{T}\left(\bar{x},t\right)}{\theta_{T,V}\left(\bar{x},t\right)}\right)\right)\right)}}\\ & \overset{Logistic\;Growth/Angiogenesis}{\overset{︷}{+k_{p,V}\phi_{V}\left(\bar{x},t\right)\left(1-\phi_{V}\left(\bar{x},t\right)/\theta_{V}\right)d}}-\overset{Death}{\overset{︷}{k_{d,V}\phi_{V}\left(\bar{x},t\right)\left(1-d\right)}} \end{array}$$
where $D_{V}\left(\bar{x},t\right)$ is the mechanically coupled vascular diffusion coefficient (described in a similar fashion as Equation (2)), *k_p,V_* is the tumor vasculature growth rate, $\theta_{V}$ is the blood volume fraction carrying capacity (assigned as the maximum observed blood volume fraction), *d* is a normalized parameter describing the distance to the periphery of the tumor (1 for voxels at the periphery, 0 for voxels furthest from the periphery), and *k_d,V_* is the vascular death rate. We note that occupancy by tumor and blood volume fractions are considered in the cross-diffusion terms, but not explicitly in the logistic growth terms in Equations (1) and (5). There are two main reasons for this choice. First, the blood and tumor volume fractions have different maximum carrying capacities. By not considering the different maximum carrying capacities it is possible to simulate voxels that are completely vascularized (which is not observed in these tumors) and would, therefore, inhibit further growth of tumor cells within that voxel. Second, we assume the blood volume fraction to dynamically update the carrying capacity of the tumor cells (via Equation (4)). In general, the carrying capacities should be interpreted as the limitations in the number of cells that a voxel can support based on the available resources (e.g., oxygen, glucose) and space. While a decrease in vascularization would allow more space for tumor cells to grow, there would be insufficient resources to support that growth thereby resulting in a decrease of the carrying capacity.

Equations (1)–(5) provide a coupled set of partial different equations that describe the spatio-temporal development of tumor cells and vasculature in the absence of RT. The following sections describe how we generate a family of models that incorporate the effects of RT in different ways.

#### 2.1.2. Mathematical Descriptions of Response to Fractionated Therapy

We developed a set of three models (*RTM*_1_ through *RTM*_3_) that are used to describe the response of tumor and vasculature to fractionated radiation therapy. The first model (*RTM*_1_) assumes there is a surviving fraction (*SF*) of tumor cells that remains following RT described by: (6)$$\phi_{T,post-RT}=\phi_{T,pre-RT}\left(C_{i}\cdot SF\right)$$
where *ϕ_T,post-RT_* is the tumor cell volume fraction immediately after RT, *ϕ_T,pre-RT_* is the tumor cell volume fraction immediately before RT, *SF* is the surviving fraction of tumor cells (between 0 and 1), and *C_i_* is one of three coupling approaches discussed below. *RTM_1_* is similar to how the LQ model has been used in other modeling approaches [13]; however, here we do not assume radiosensitivity parameters (e.g., *α* or *β*) a priori and simply fit (for each animal) the effect of RT as the *SF*. During simulation time steps that correspond to RT treatment times, Equation (6) is evaluated prior to solving Equations (1)–(5). The post-RT tumor volume fraction, *ϕ_T,post-RT_*, is then used as the current estimate of the tumor distribution used in the finite difference approximation of Equations (1)–(5). *RTM_2_* includes the death term from *RTM_1_*, while also reducing *k_p,T_*, in Equation (1), with each additional fraction of RT:(7)$$k_{p,T}=k_{p,T,0}\left(C_{i}\cdot SF^{n}\right)$$
where *k_p,T,_*_0_ is the pre-treatment proliferation rate, and *SF^n^* represents the fraction of tumor cells that are able to proliferate following *n* fractions of RT. *RTM_3_* removes the assumption of instantaneous death (i.e., Equation (6)), and instead assumes tumor cells die at a rate *k_d,T_*: (8)$$k_{d,T}=k_{d,T,0}\left(1-C_{i}\cdot SF^{n}\right)$$
where *k_d,T,_*_0_ is the death rate due to RT, and $\left(1-C_{i}\cdot SF^{n}\right)$ is the fraction of dying cells after *n* fractions of RT. *RTM_3_* amends Equation (1), by adding an exponential death term shown below: (9)$$\begin{array}{ll} \frac{\partial \phi_{T}\left(\bar{x},t\right)}{\partial t}= & \overset{Diffusion}{\overset{︷}{\nabla \cdot \left(D_{T}\left(\bar{x},t\right)\cdot \left(\left(\left(1-\frac{\phi_{V}\left(\bar{x},t\right)}{\theta_{T,V}\left(\bar{x},t\right)}\right)\nabla \frac{\phi_{T}\left(\bar{x},t\right)}{\theta_{T,V}\left(\bar{x},t\right)}\right)+\left(\frac{\phi_{T}\left(\bar{x},t\right)}{\theta_{T,V}\left(\bar{x},t\right)}\nabla \frac{\phi_{V}\left(\bar{x},t\right)}{\theta_{T,V}\left(\bar{x},t\right)}\right)\right)\right)}}\\ & +\overset{Logistic\;Growth}{\overset{︷}{k_{p,T}\phi_{T}\left(\bar{x},t\right)\left(1-\phi_{T}\left(\bar{x},t\right)/\theta_{T}\left(\bar{x},t\right)\right)}}-\overset{Death\;due\;to\;RT}{\overset{︷}{k_{d,T}\phi_{T}\left(\bar{x},t\right)}}\;\; \end{array}$$

Identical formulations of Equations (6)–(8) are used to model the effect of RT on the vasculature. In addition to these three models (i.e., Equations (6)–(8)) of radiation induced cell death, we also developed three ways to spatially vary the efficacy of RT through the coupling coefficient *C_i_* [9]. The first coupling approach (*C*_1_) maximizes the *SF* as *ϕ_T_* approaches *θ_T_* due to an assumed slower tumor cell proliferation (thereby making the cells less susceptible to RT):(10)$$C_{1}\left(\bar{x},t\right)=\phi_{T}\left(\bar{x},t\right)/\theta_{T}\left(\bar{x},t\right)$$

The second coupling approach relates the efficacy of RT to *ϕ_V_ via*: (11)$$C_{2}=\exp\left(-\alpha_{1}\left(\phi_{V}\left(\bar{x},t\right)/\theta_{V}\right)\right)$$
where *α*_1_ is a coupling constant. Thus, regions with high *ϕ_V_* experience an increased treatment efficacy (due to assumed increased oxygenation). For the third coupling approach (*C*_3_), we assume *C*_3_ is equal to 1 everywhere. Combining the three models of response to RT (Equations (6)–(8)) and the three approaches to spatially vary response (i.e., *C*_1_–*C*_3_) we have a total of nine models of response to radiation therapy. 

### 2.2. Experimental Methods

#### 2.2.1. Animal Model and Radiation Therapy Protocol

All experimental procedures were approved by our Institutional Animal Care and Use Committee. Seven female athymic Hsd:RH-*Foxn1^RNU^* nude rats (weighing from 186 to 220 g) were purchased from Envigo (Indianapolis, IN, USA). The human glioma cell line U-87 MG was purchased from ATCC (ATCC HTB-14, Manassas, VA, USA) and were grown as per the packaging. U87-MG cells were cultured to 80% confluence (1–2 weeks) in Eagle’s Minimal Essential Media (ATCC 30-2003) supplemented with 10% FBS (A3160502, Thermo Fisher, Waltham, MA, USA), 100U/mL penicillin-streptomycin (15-140-122, Fisher Scientific, Houston, TX, USA), 500 ng/mL amphotericin B (BS721, BioBasic, Amherst, NY, USA), 500 ng/mL plasmocin mycoplasma prophylactic (ant-mpp, Invivogen, San Diego, CA, USA). Cells were then trypsinized, washed 2×, and resuspended in 1 × PBS at a concentration of 8 × 10^7^/mL. For tumor cell injections [31], the animals were anesthetized (2% isoflurane in 100% oxygen) and the needle was positioned 1 mm posterior and 2 mm right lateral to the Anterior-Posterior (AP) and Medial-Lateral (ML) coordinates of bregma, respectively, then advanced 4 mm ventral (deep) to skull surface. The needle was then lowered into the skull slowly over 2 min, then allowed to remain in situ another 2 min prior to injection. Injection was carried out at a rate of 0.4 µL/min over approximately 6 min. After injection, pressure was allowed to equalize with the needle in situ for 10 min prior to retraction. A total of 2 × 10^5^ U-87 MG glioma cells in a 2.5 µL volume. 48 h prior to their first MRI study, a permanent jugular catheter was placed in each rat.

Imaging studies typically began when the tumor volume reached 30 to 50 µm^3^. Three days later, animals were randomly assigned to either the 2 Gy or 4 Gy per fraction treatment group. Fractionated RT was delivered five days per week for two consecutive weeks. Whole brain RT was delivered using the MultiRad350 (350 kVp/4.7 mA, Precision X-ray Irradiation, North Branford, CT, USA). During the irradiation procedure, the animals were anesthetized with a mixture of 2% isoflurane in 100% oxygen, and then positioned within the irradiation chamber in a prone position with the animal’s head placed at the center of the irradiation platform. Lead blocks were placed to shield the animal’s torso. RT was delivered at a dose rate of 1.0 Gy/min in the dorsal to ventral direction, and total dose delivered was controlled by setting the exposure time to either 2 min or 4 min. The RT beam was filtered using a Sn-Al-Cu filter (Precision X-ray Irradiation). The dose rate and planned total dose for both the 2 min and 4 min exposure was verified daily using the MultiRad 350’s integrated dosimeter prior to animal irradiation. Imaging continued up to three times per week during and after the completion of RT. We do note, however, that the imaging and RT experimental time line varied between animals due to animal health concerns (e.g., excessive weight loss or dehydration, tumor growth impacting mobility) that required them to be euthanized prior to the end of the study. Table 2 reports the exact imaging and RT time line for each animal.

#### 2.2.2. Imaging Procedure

Multiparametric MRI was acquired using a 7.0T horizontal-bore magnet (Bruker Biospec, Billerica, MA, USA) with a 60 mm diameter volume coil over a 32 × 32 × 16 mm^3^ field of view. Multiparametric MRI consisted of inversion recovery data to construct a *T*_1_-map, *T*_2_-weighted MRI for anatomical images, diffusion-weighted (DW-) MRI data to compute the apparent diffusion coefficient (ADC), and dynamic contrast-enhanced (DCE-) MRI data to compute the blood volume fraction. All images were acquired with a 128 × 128 matrix and 16 slices. 

Data for the pre-contrast *T*_1_ map were acquired using a segmented FLASH (segFLASH) inversion recovery sequence with: *α* = 15°, *TE* = 3.2 ms, *segment size* = 12. The longitudinal relaxation curve was sampled at 30 inversion times (*TI*) ranging from 125 to 3025 with a 100 ms spacing. Voxel-wise *T_1_* values were estimated by fitting Equation (12) to each voxel’s relaxation curve [32]:(12)$$\begin{array}{c} S(TI)=A-B\exp\left(-TI/T_{1}^{*}\right)\\ A=S_{0}T_{1}^{*}/T_{1}\\ B=S_{0}\left(1+T_{1}^{*}/T_{1}\right)\\ T_{1}=T_{1}^{*}\left(\frac{B}{A}-1\right) \end{array}$$
where $T_{1}^{*}$ is the effective T_1_, and *S*_0_ is the inherent signal intensity. *A*, *B* and $T_{1}^{*}$ were estimated using a non-linear least square optimization (*lsqcurvefit* in MATLAB, Mathworks, Natick, MA, USA) and then used to calculate *T*_1_. 

*T*_2_-weighted MRI data was acquired using a fast spin-echo or rapid imaging with refocused echoes (RARE) sequence with the following pulse sequence parameters: *TR* = 3500 ms, *TE* = 14 ms, RARE factor of 8, and number of excitations = 10.

DW-MRI data was acquired using a pulsed gradient echo sequence with three *b*-values (150, 350, and 800 s/mm^2^) and gradients applied simultaneously along the three orthogonal directions with the following pulse sequence parameters: *TR* = 2500 ms, *TE* = 28.7 ms, number of excitations = 2, *Δ*
*=* 23 ms, and *δ* = 3 ms. The DW-MRI data was modeled by a standard mono-exponential decay to estimate the apparent diffusion coefficient (ADC) voxel-wise within the tumor. The tumor cell volume fraction, $\phi_{T}\left(\bar{x},t\right)$, was then estimated directly from the *ADC* map using Equation (13):(13)$$\phi_{T}\left(\bar{x},t\right)=\left(\frac{ADC_{w}-ADC\left(\bar{x},t\right)}{ADC_{w}-ADC_{\min}}\right)$$
where *ADC_w_* is the ADC of free water at 37 °C [33], $ADC\left(\bar{x},t\right)$ is the ADC value at position $\bar{x}$ and time *t*, and *ADC*_min_ is the minimum ADC observed within the tumor regions-of-interest (ROIs) across all animals. We note that while we have extensively used the ADC to estimate tumor cellularity [9,21,25], we acknowledge that there are other factors that may influence the measured ADC (e.g., cell size and permeability). This point is discussed further in [21].

DCE-MRI data was collected using a *T*_1_-weighted spoiled gradient echo sequence with *TR =* 101 ms, *TE* = 1.9 ms, and a flip angle of 22°. A 200 µL bolus (0.05 mmol kg^−1^) of Gado-DTPA^TM^ (BioPhysics Assay Lab, Worcester, MA, USA) was injected after 25 image volumes were acquired. The relative blood volume fraction, $\phi_{V}(\bar{x},t)$, was calculated by computing the ratio of the area under the curve for the concentration of the contrast agent time course for each tissue voxel to the area under the arterial input function [34] over the first 60 s [9,21].

Following the first imaging session, a mutual information based rigid registration algorithm was used at the beginning of each subsequent imaging session to register the current animal placement to the first imaging session placement [25]. The resulting spatial offsets and rotations were applied when selecting the field of view on the console to minimize the need for post-acquisition registration.

After all imaging studies were completed for an individual rat, the same registration algorithm was employed (as needed) to maximally align the imaging data across time. The rigid registration transformation was estimated using *imregtform* in MATLAB R2020a using the *multimodal* configuration which employs a mutual information-based cost function. Registration performance was visually assessed by overlaying the registered and target (i.e., initial image) over the central eight slices. If the initial automatic registration performance was inadequate, a manual transformation was applied as an initial transformation for *imregtform*. Tumor regions of interest were segmented using a semi-automated approach by an imaging scientist with over ten years of experience in segmenting contrast-enhancing lesions in murine models of glioma. The segmented tumor consisted of the enhancing lesion on a post-contrast *T*_1_-weighted MRI (from the DCE-MRI dataset). First, a region of interest is manually drawn around the contrast-enhancing lesion. Second, a k-means clustering in MATLAB R2020a is used to identify voxels that are enhancing and non-enhancing within the region of interest. Third, *imfill* in MATLAB R2020a is used to fill in holes within the regions identified as enhancing tissue. Finally, the k-means segmented tumor is visually inspected before proceeding to modeling. The robustness of this approach was evaluated in an in silico study where noise was added from a normal distribution (equivalent to an SNR of 20) to each animal’s day 0 post-contrast *T_1_*-weighted MRI to generate 100 unique imaging volumes which were then segmented using the semi-automated approach just described. We then calculated the variability in volume estimates and the degree of spatial overlap using the standard error and Dice correlation coefficient, respectively. We observed that this semi-automated approach is robust to the noise level observed in the image used for segmentation resulting in a standard error of less than 0.26 mm^3^ and Dice correlation coefficients greater than 0.91 for all animals. Complete results are reported in the Appendix A. The first panel in Figure 1 provides a schematic of our image processing and modeling approach. 

### 2.3. Numerical and Computational Methods

#### 2.3.1. Finite Difference Approximation

While complete numerical details are described elsewhere [30], here we present the salient details. A finite difference approximation implemented in MATLAB R2020a was used to determine the spatial-temporal evolution of $\phi_{T}\left(\bar{x},t\right)$ and $\phi_{V}\left(\bar{x},t\right)$ using a fully explicit in time differentiation (time step = 0.01 days) and three dimensions in space (Δ*x* = 250 µm, Δ*y* = 250 µm, Δ*z* = 1000 µm) central difference spatial differentiation. No flux boundary conditions were applied at the skull boundary for $\phi_{T}\left(\bar{x},t\right)$ and $\phi_{V}\left(\bar{x},t\right)$ at the skull boundary. The boundary condition for $\vec{u}$ was assumed to be zero displacement in the normal direction, while it was assumed that the tissue in the tangential directions was free to move (i.e., slip condition). 

#### 2.3.2. Parameter Calibration (Scenario 1) and Tumor Response Forecasting (Scenario 2)

As outlined in Section 2.1, we developed a family of 18 models consisting of nine approaches accounting for the effects of RT (three models of RT response (*RTM_1_*-*RTM_3_*) and three approaches (*C*_1_–*C*_3_) to spatially vary the effect of RT), and two approaches to calibrate *k_p,T_*. The model parameters listed in Table 1 were calibrated (second panel in Figure 1) for all 18 models and each animal using a hybrid simulated-annealing Levenberg-Marquardt algorithm [21]. All parameters were considered to be global (uniform throughout the domain) with the exception of *k_p,T_*, which was calibrated either as a global variable or varied spatially throughout the domain. When *k_p,T_* was defined as a field, parameter values were only calibrated within a subset of points within the tumor and then interpolated elsewhere to reduce the number of individual parameters needed to be calibrated. More specifically, for a given 3 × 3 region of voxels within the tumor, the parameter values were calibrated at the corner and center positions while the remaining four points were interpolated from the nearest calibrated values. This calibration approach also regularizes (or smooths) the parameter field spatially. (Bounds used for model calibration are reported in Appendix A). We performed two calibration scenarios to determine how well the model describes the data and how well it forecasts future response. For scenario 1, we calibrated each model to all imaging days for an individual animal to evaluate how well each model describes the data. Once we have calibrated the model parameters, we then determined the parameters 95% confidence intervals (third panel in Figure 1) using *nlparci* in MATLAB. We then sampled the parameter confidence intervals to generate 100 sets of model parameters that were then used in 100 additional finite difference forward evaluations to estimate tumor growth and response. For each of the 100 tumor growth calculations, we assessed the error as described below to generate confidence intervals in our model output. For scenario 2, we seek to determine the predictive strength of these models. To do so, we first calibrated each model to the first half of the available data for each animal. We then (similarly to scenario 1) sampled the confidence intervals of the calibrated parameters to generate a set of 100 model parameters that were then used in 100 additional finite difference forecast of tumor growth and treatment response that was directly compared to the last half of the available data for each animal. For each of these 100 forecasts, we also determined the error as described below to generate confidence intervals in our predictions.

#### 2.3.3. Model Selection and Ensemble Average

The Akaike Information Criterion (AIC) [35] was used to select the model that optimally balanced model complexity and agreement with the data. The AIC is defined as:(14)$$AIC=2k+n\ln\left(\frac{RSS}{n}\right)+2k\left(\frac{k+1}{n-k-1}\right)$$
where *k* is equal to the number of calibrated parameters for each model, *n* is the number of data points used to calibrate the model, and *RSS* is the residual sum squares between the model and measured tumor and blood volume fractions calculated over the total data used for calibration. For each animal in the first calibration scenario (i.e., the scenario where all models are calibrated to the entire time-course of imaging data obtained for each animal), we calculated the AIC for each model. We then report results for the model with the lowest AIC as well as the ensemble average model. The AIC for each model was used to calculate the ensemble average weights defined as:(15)$$w_{i}=\frac{\exp\left(-\frac{\delta_{i}}{2}\right)}{\sum_{j=1}^{18}\exp\left(-\frac{\delta_{j}}{2}\right)}$$
where *w_i_* is the weight for the *i*-th model, $\delta_{i}$ is equal to $AIC_{i}-AIC_{\min}$, and *AIC*_min_ is the minimum observed AIC. The ensemble averaged tumor volume fraction was calculated using:(16)$$\phi_{T,ens}=\sum_{j=1}^{100}\sum_{i=1}^{18}w_{i}\phi_{T,i,j}$$
where $\phi_{T,ens}$ is the ensemble average, and $\phi_{T,i,j}$ is tumor volume fraction for the *i*-th model and the *j*-th set of parameters sampled from the parameter confidence intervals (described in Section 2.3.2). The ensemble averaged blood volume fraction was calculated in a similar way as Equation (16). For the second calibration scenario (i.e., the prediction scenario), we report results using the model with the lowest AIC (calculated for each animal during the second calibration scenario) as well as the ensemble average of the model forecasts. 

### 2.4. Error Quantification

The model calculated tumor and blood volume fractions were compared directly to the measured values (see the fourth panel of Figure 1). At the global level, we calculated the percent error in predicted tumor volume and the Dice coefficient (describing the degree of overlap of the measured and model calculated volumes). At the local level we calculated the Pearson correlation coefficient (PCC) and the concordance correlation coefficient (CCC), which characterize the degree of correlation and agreement, respectively, between the model and the measurement at each voxel location. For each model calculated tumor and blood volume fractions derived from each set of model parameters (sampled from the parameter confidence interval), we assessed the global and local level errors at each time point. Each of the above-mentioned error metrics were then averaged for each set of model parameters across time. That is, for each model and animal we have 100 estimates of PCC, CCC, Dice, and percent error in tumor volume. The results for each animal are reported as box and whisker plots. 

## 3. Results

### 3.1. Model Calibration Scenario

Figure 2 and Figure 3 report the global and local-level error analysis, respectively, for the first calibration scenario. The top panel in Figure 2 shows the model weights for each individual animal as well as the average weight across animals. In general, models with a voxel-specific *k_p,T_* (models 1–9) were weighted more heavily (contributing 80.7% of the ensemble weight) compared to the global *k_p,T_*. Additionally, the models that incorporated a death rate (rather than instantaneous death) were weighted the most (models 7–9) and contributed 56.9% of the ensemble weight. 

At the global level (bottom two panels of Figure 2), the selected model (orange box plots) had median percent errors in tumor volume predictions less than 8.7%, while the ensemble averaged model had percent errors in tumor volume predictions less than 18.1%. A high level of spatial overlap was observed across all animals resulting in Dice values greater than 0.68. Six out of seven animals had statistically significant differences (*p* < 0.05) between the selected model and the ensemble average for both the percent error in tumor volume and Dice values. 

Figure 3 reports the local-level error analysis for both tumor (left column) and blood volume estimates (right column). At the local level for both the selected and ensemble average model (Figure 3), we observed a strong degree of correlation (PCCs > 0.74) for all animals between the observed and model estimated values of tumor volume fraction. Similarly, a high degree of agreement (CCCs > 0.76) was observed for animals 2–4, while the remaining animals, comparatively, had a lower degree of agreement (CCCs > 0.53). All seven animals had statistically significant differences (*p* < 0.05) between the selected and ensemble average model. 

For blood volume estimates, a high degree of correlation (PCCs > 0.70) was observed for the selected model for all animals, while lower correlation was observed for the ensemble average for animals 5 and 6 (PCCs < 0.56). CCCs greater than 0.50 were observed for the selected model estimate of blood volume fraction for all but animal 5. The ensemble average generally resulted in lower agreement (CCCs > 0.14) compared to the selected model. Statistically significant differences (*p* < 0.05) were observed between the selected and ensemble model for six out of seven animals. Calibrated model parameters for each animal are reported in Table 3 for the selected model. 

### 3.2. Model Prediction Scenario

Figure 4 and Figure 5 report the global and local error analysis, respectively, for the prediction scenario for animals 1, 3, 5, 6, and 7. The top panel in Figure 4 shows the model weights for each individual animal as well as the average weight across animals. Similar to the first calibration scenario, models with a voxel-specific *k_p,T_* (models 1–9) were weighted more heavily (contributing 77.5% of the ensemble weight) compared to the global *k_p,T_*. Likewise, models that incorporated a death rate (rather than instantaneous death) were weighted the most (models 7–9) and contributed to 54.9% of the ensemble weight. 

In general, a low error (less than 16.2%) was observed in tumor volume predictions across all animals for the selected model. The ensemble average model resulted in significantly higher error (*p* < 0.05) for all animals. Additionally, the Dice correlation coefficient was greater than 0.60 for the selected model. The ensemble average model resulted in significantly lower (*p* < 0.05) Dice correlation coefficients for four of the animals. 

Predictions of the tumor volume fraction at the local level (left column in Figure 5) resulted in strong correlation for animals 1 and 3 (PCCs > 0.80) and lower correlation for the remaining animals (PCCs > 0.57) for the selected model. The ensemble average model resulted in PCCs greater than 0.69 for all animals. Reduced agreement compared to the calibration scenario was observed resulting in CCCs greater than 0.58 for animals 1, 3, and 7 for both the selected and ensemble average model. The ensemble average model resulted in lower agreement compared to the selected model for three out of five animals. 

For blood volume fraction predictions at the local level (right column in Figure 5), the selected and ensemble average models both resulted in strong correlation (PCCs > 0.65). Lower agreement was observed across the cohort resulting in a median CCC greater than 0.49 for the selected model and 0.36 for the ensemble average model. 

Figure 6 and Figure 7 display the results for the tumor growth and response predictions. Figure 6 and Figure 7 display the central slice predictions of tumor and blood volume fraction, respectively, for all five animals. Figure 6 shows the results for animals 1 and 3, while Figure 7 shows the results for animals 5 through 7. Sagittal views of Figure 6 and Figure 7 are shown in Appendix A. Figure 8 details the tumor volume predictions overtime for both the selected model and ensemble average for each animal presented in Figure 6 and Figure 7. Time-resolved errors in tumor volume and the Dice correlation coefficient are shown in Appendix A. For animal 1, a high level of voxel-wise agreement was observed for tumor volume predictions (PCC = 0.86 and CCC = 0.80) compared to the blood volume predictions (PCC = 0.70 and CCC = 0.42) for the selected model. Notably, both models were able to predict the area of low cell density in the first four prediction time points. For animal 3, the model tends to underestimate the tumor area in the central slice; however, a high level of agreement is observed at the voxel level (PCC > 0.84 and CCC > 0.72) for both the selected and ensemble model. For animal 5, we observed a median error less than 8.7% for both models in tumor volume predictions, and strong (PCC = 0.93 and CCC = 0.74) voxel level agreement for the ensemble average model. Animal 6, had predictions over 11 days which resulted in a median error of 16.0% in tumor volume predictions for the model with the lowest AIC, while the ensemble average model had greater than 100% error in tumor predictions. For animal 7, we predicted tumor growth from days 10 to 17, which resulted in a median error less than 21.5% for both models in tumor volume predictions. Additionally, both models tended to overestimate blood volume predictions at day 14 and 17 resulting in CCCs less than 0.26. 

## 4. Discussion

We have developed and applied an experimental-computational framework to predict the response of murine tumors to fractionated radiation on an individual subject basis. More specifically, we parameterized 18 biologically-based models of tumor and vasculature response to fractionated RT via quantitative MRI data obtained in seven animals, and then used the calibrated models to predict the spatio-temporal development of key tumor characteristics. Non-invasive imaging data from DW- and DCE-MRI enabled estimates of tumor and blood volume fractions that served as ground truth for model calibration, selection, and evaluation of prediction accuracy. We evaluated two calibration scenarios to assess how well these models: (1) describe the entire tumor growth time course, as well as how well they (2) predict the remaining imaging visits when half of the data is used for calibration. For the first scenario, when calibrated to all available imaging time points, the model with the lowest AIC resulted in less than 8.7% error in tumor volume estimates across all animals. While the ensemble average model resulted in less than 18.1% error in tumor volume estimates across all animals for the first scenario. Notably, the ensemble average model resulted in a decrease in tumor volume error for three out of seven animals, and an increase in CCCs for all seven animals. For the second scenario, when we calibrate over the first half of the time points and predict the remaining time points, a median error of less than 16.2% in tumor volume estimates were observed across all animals for the model with lowest AIC. Unlike the first calibration scenario, the ensemble average prediction resulted in increased tumor volume estimates compared to the model predictions with the selected model. However, increased correlation (PCCs) and agreement (CCCs) were observed for three out of the five and two out of the five animals, respectively, for the ensemble average prediction. We note, that we did not observe significantly different response between treatment groups. Notably animals 1, 6, 7 were all imaged at least 5 days post RT despite receiving different treatment doses. We hypothesize the varied response could be due to variations in tumor growth properties or sensitivity to RT itself. For example, animal 1 presents with the lowest growth rate (Table 3) of the animals, had the longest overall survival, and received only 2 Gy/day. This result may provide further motivation for personalizing RT regimens to adapt to individual tumor properties. This preliminary study indicates a promising approach for personalizing mathematical models of response to fractionated RT.

Our approach extends our previously developed image-driven modeling framework applied in the presence [21,25] and absence of radiotherapy [9] in the C6 glioma line. In the present effort, we refined our experimental approach in three main ways to improve upon previous studies. First, we applied our experimental-computational framework to the U-87 cell line. Second, animals received smaller radiation fractions (2 or 4 Gy) instead of single large dose (20 or 40 Gy) to more closely mimic how RT is delivered clinically. While the delivery of RT is still substantially different (whole brain versus highly conformal RT beams), this approach facilitated the study of temporal response to RT that is not possible through single fraction of RT. Third, we constructed for each animal an ensemble average model weighted using the AIC. In general, the ensemble average model did not outperform the selected model, but by weighting model outcomes we have a forecast that considers all of the possible tumor response patterns. In addition, the model weights themselves provide some insight into the subject-specific growth and response characteristics. We investigated ensemble averages (and individual model simulations) as they could deliver a powerful tool for clinicians providing a forecast of the average, best, and worst response scenarios for an individual subject at the beginning of therapy. Similar to weather forecasting [36], as additional data is observed model weights could be adjusted based on forecast performance to provide an updated ensemble forecast. 

Accurate characterizations and forecasts of the temporal response of tumors to radiation therapy are critical to the development of patient optimized treatment regimens. Optimized radiotherapy regimens may be able to address variations in tumor properties (e.g., hypoxia) that alter radiosensitivity and influence response to radiotherapy [37]. Several promising computational oncology studies have investigated applying mathematical modeling to systematically evaluate alternative regimens [17,38,39]. These approaches focus primarily on adapting or optimizing regimens based on changes in tumor geometry. However, we posit that these approaches could be refined further to adapt or optimize RT regimens based on both tumor geometry and intratumor radiosensitivity. By reducing tumor biology to simply the shape and location of the tumor, we are blind to spatial variations in tumor response that might ultimately lead to disease progression. Our coupled model of tumor growth and angiogenesis forecasts a spatial map of response that could be targeted by intensity modulated RT. The results in this preliminary cohort indicate strong predictive accuracy in tumor geometry (low error in tumor volume predictions) while additional development is needed to improve local level predictions. Our current model of response to RT is relatively simple compared to the complex biophysical mechanisms of response to radiotherapy, which are undoubtedly varying spatially and temporal during fractionated RT. However, additional experimental data is required to properly initialize and constrain a more complete description of tumor radiobiology. In the clinical setting, one potential application for this modeling framework is to predict long term response (or time to progression) to guide alternative fractionation schemes with the end goal of improving patient outcomes [17,28]. While short-term predictions performed well, further development may be needed to improve long term predictions. Once a model is established that can accurately reproduce the spatiotemporal development and response of the tumor it can then be used to identify alterative treatment regimens or fractionation schemes that may outperform standard methods. 

With regards to calibrated parameter values, we observed that the calibrated *k_p,T,_*_0_ and *D_T,_*_0_ are within ranges reported in literature for image-based estimates of proliferation and diffusion [25,40] except for animal 6. For animal 6, *k_p,T,_*_0_ and *D*_*T*,0_ were at least 2.3 and 1.5 times higher, respectively, than the other animals. The larger growth rates for animal 6 might contribute to the higher errors observed in both the calibration and prediction scenarios. The poorer prediction may be due to the rapid or aggressive tumor early on that results in higher proliferation and diffusion coefficients. Individual tumor forecasts could be compared to a population averaged forecast to identify subjects whose tumor growth or response forecasts deviate significantly from the population. Additionally, we observed a dose-dependency on *SF* where animals that received 2 Gy was higher (*SF* > 0.95) than those animals that received 4 Gy (*SF* < 0.91).

There are several opportunities for further investigation and development of our experimental-computational approach. First, we assume that the measured ADC can provide reasonable estimates of tumor volume fraction. As we have previously discussed [21], the ADC is influenced by a combination of both cellular and tissue properties including cell density, cell size, and cell permeability. However, in our formulation, we assume the changes observed in the ADC over time are influenced predominately by cell density. Advanced diffusion techniques [41] may be able to probe these properties further and provide a more complete description of tumor tissue and cell properties. Second, while the present data set does provide evidence that personalizing mathematical models of response to fractionated RT is a promising avenue to investigate, further experimental studies are needed to increase the cohort size and provide a more complete characterization of the predictive accuracy of these models. Third, future studies should consider including additional tumor cell lines that are more infiltrative and better recapitulate human glioblastoma characteristics, thereby testing the generalizability of the experimental-computational approach. While the U-87 line, is a human-derived glioblastoma line, it may fail to capture some of the key characteristics of the human disease [42]. Notably, U-87 tumors tend not to be diffuse or infiltrative and remain well circumscribed and vascularized. As such, the U-87 line enables reliable estimation of tumor burden (and assignment of ground truth) making it suitable for model development and refinement. However, alternative cell lines that better recapitulate human glioblastoma should be considered for further development of this technique. Other human derived cell lines which have a more invasive pathology (e.g., the U-251) should be considered to evaluate the generalizability of this approach to different physio-pathological conditions. One challenge with any of these human derived cells is the use of immunocompromised animals which may not accurately recapitulate the interactions between the tumor and host, or the immune response to RT [43]. Fourth, genetic and phenotypic diversity [44] is a significant factor in tumor growth and response of tumors to systemic therapy and radiotherapy. Although this diversity is not explicitly described in our mathematical model, we hypothesize that these variations may be implicitly captured through the individualized model parameterization. To some extent, phenotypic diversity may be accounted for through calibrating a proliferation field rather than assuming homogenous tumor growth properties. To test this hypothesis, tumor growth predictions should be compared to other approaches that explicitly integrate genetic and phenotypic diversity with mechanism-based models [45]. Fifth, future studies, in larger cohorts, are needed to investigate whether it is important to perform subject-specific model selection or use an ensemble average model whose weights are determined from a training set. Sixth, future efforts should consider more complete descriptions of tumor-induced angiogenesis and regression. Our model of tumor-induced angiogenesis and regression is an over-simplification of the complex mechanisms of vasculature creation and destruction observed in vivo [46]. As such, it may fail to accurately capture angiogenesis in all tumors. Finally, in this modeling approach we assumed the delivered dose to be uniform throughout the tumor, which may not be the case for a single angle delivery of RT. Variability in the delivered dose should be considered to more accurately capture response dynamics spatially. In the clinical setting, RT treatment plans could be used to provide a spatial map of the planned RT throughout the brain.

## 5. Conclusions

We have developed and applied a novel image-driven and biologically-based modeling framework for characterizing and forecasting response of both tumor and vasculature tissue to fractionated radiation therapy. Preliminary results indicate low error in characterizing and predicting tumor volume and local cell number. Thus, further investigation is warranted, and future efforts should apply this approach to a larger cohort and a broader range of fractionation schemes. 

## Figures and Tables

**Figure 1 cancers-13-01765-f001:**
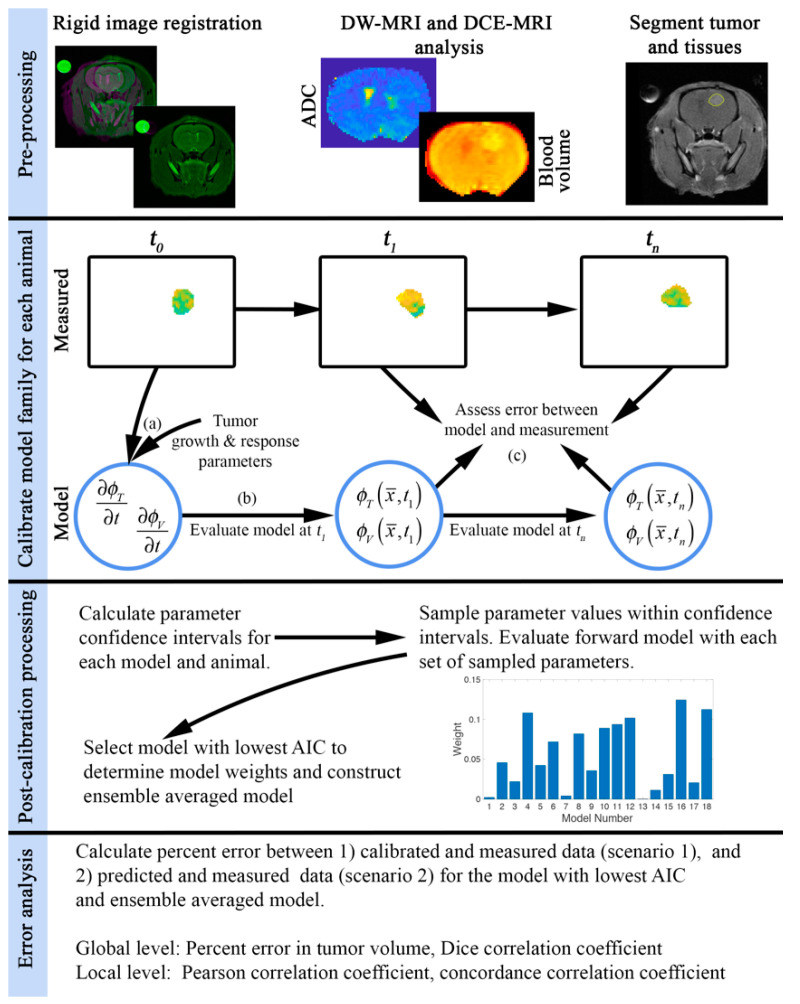
Overview of the image processing and computational approach. Pre-processing: Images are first aligned to the first imaging session using a rigid registration. We then analyze DW-MRI and DCE-MRI to estimate the ADC and blood volume, respectively. Contrast-enhanced images are then used for the segmentation of tumor burden. Calibrate model family for each animal: Each model within the model family is calibrated for each individual animal. Tumor and blood volume fractions at baseline (*t*_0_) and an initial guess of model parameters, (a), are used to initialize our 3D model of tumor growth and response. The mathematical model is evaluated at subsequent imaging days *t*_1_ to *t_n_*, (b), and the error between the model and the measured tumor and blood volume fractions are assessed and are used to refine the estimates of model parameters (c). Post-calibration processing: Confidence intervals for each set of model parameters are determined for each model and animal. We then sample the parameter confidence intervals and evaluate the model with each set of sampled parameters to assess the variations in model outcomes. The Akaike Information Criterion is then used to select the most parsimonious model and determine model weights. The ensemble averaged model is then constructed by weighting each model output. Error analysis: Error between the measured and model estimated tumor and blood volume fraction is then analyzed at the global and local levels.

**Figure 2 cancers-13-01765-f002:**
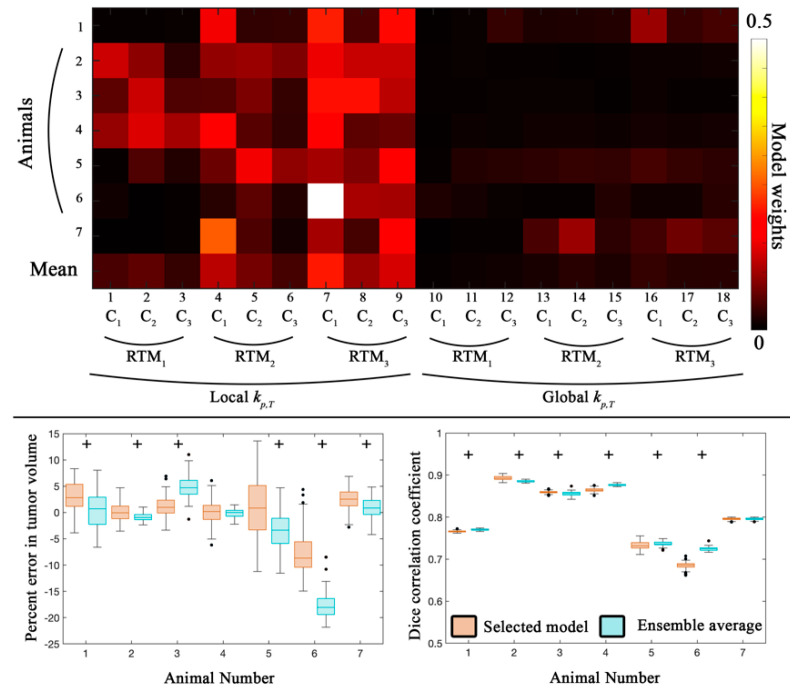
Global error analysis from the calibration scenario. The top panel shows model weights determined from the Akaike Information Criterion for all seven animals. *C*_1_ through *C*_3_ correspond to the different approaches to introduce spatial variation in tumor response, while *RTM*_1_ through *RTM*_3_ correspond to the different RT response models. Models with a local *k_p,T_* and *RTM*_3_ accounted for 56.9% of the ensemble average model. In the bottom panel, the average percent error in tumor volume and the Dice correlation coefficient are reported for the selected model (orange) and the ensemble averaged model (blue) for all seven animals. The median error for the selected model resulted in less than 9% error for all animals. The median error for the ensemble average was also less than 6% error, with the exception of animal 6 which had greater than 19% error. A strong level of spatial overlap (Dice > 0.68) was observed for all animals. Statistically significant differences (*p* < 0.05) between model estimates are indicated by the ‘+’ symbol. Model 7 was weighted the highest with an average weight of 0.21.

**Figure 3 cancers-13-01765-f003:**
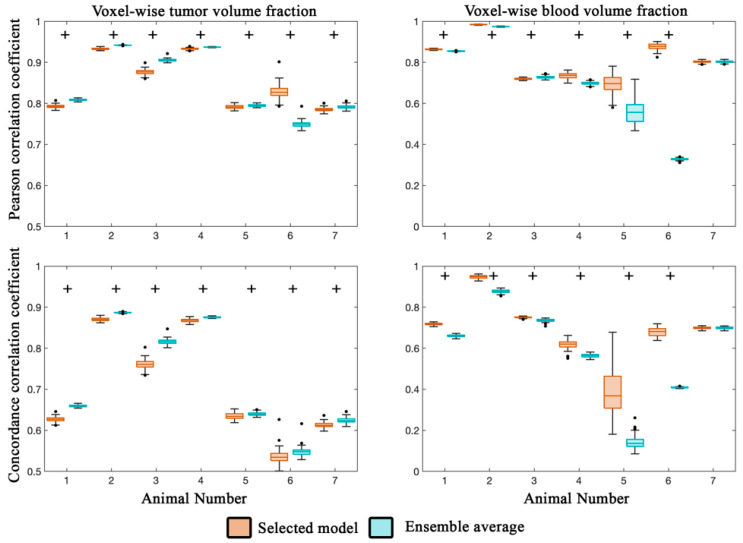
Local error analysis from the calibration scenario. The PCC (top row) and CCC (bottom row) are reported for the tumor (left column) and blood volume (right column) fractions for the calibration scenario. In general, a high level of agreement and correlation was observed for the tumor volume fraction (PCCs > 0.74 and CCCs > 0.53) for the selected and ensemble average model estimates. Similarly, a high level of correlation was observed for voxel-wise estimates of blood volume fraction (PCCs > 0.70). CCCs were greater than 0.50 for all but animal five. Statistically significant differences (*p* < 0.05) between model estimates are indicated by the ‘+’ symbol.

**Figure 4 cancers-13-01765-f004:**
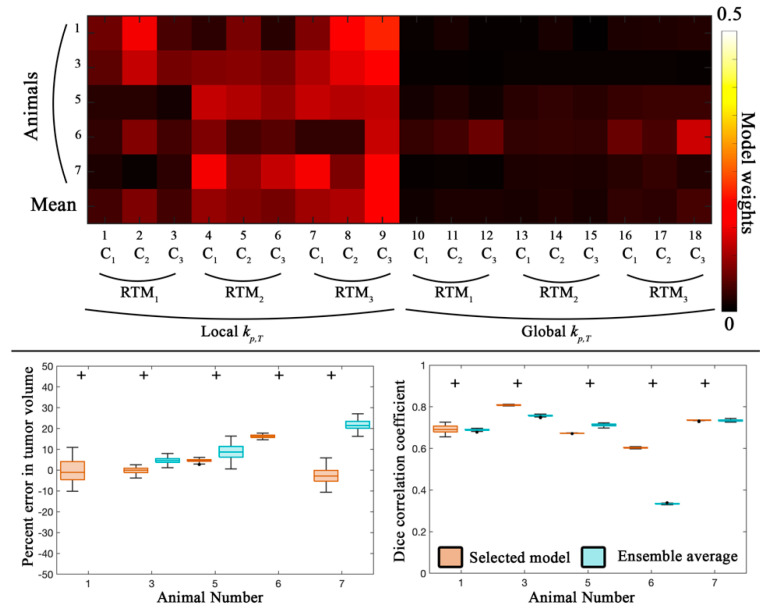
Global error analysis from the prediction scenario. The top panel shows model weights determined from the Akaike Information Criterion for the five animals from the prediction scenario. *C*_1_ through *C*_3_ correspond to the different approaches to introduce spatial variation in tumor response, while *RTM*_1_ through *RTM*_3_ correspond to the different RT response models. Models with a local *k_p,T_* and RTM_3_ accounted for 54.9% of the ensemble average model. In the bottom panel, the average percent error in tumor volume and the Dice correlation coefficient are reported for the selected model (orange) and the ensemble averaged model (blue) for five animals. (Animals 2 and 4 had insufficient imaging visits for the prediction scenario). The median error for the selected model resulted in less than 16.2% error for all animals. For animal 1 and 6, the ensemble average greatly overestimated the tumor volume with a median error of greater than 100%. A high level of spatial overlap (Dice values greater than 0.60) was observed for the selected model. Statistically significant differences (*p* < 0.05) between model estimates are indicated by the ‘+’ symbol.

**Figure 5 cancers-13-01765-f005:**
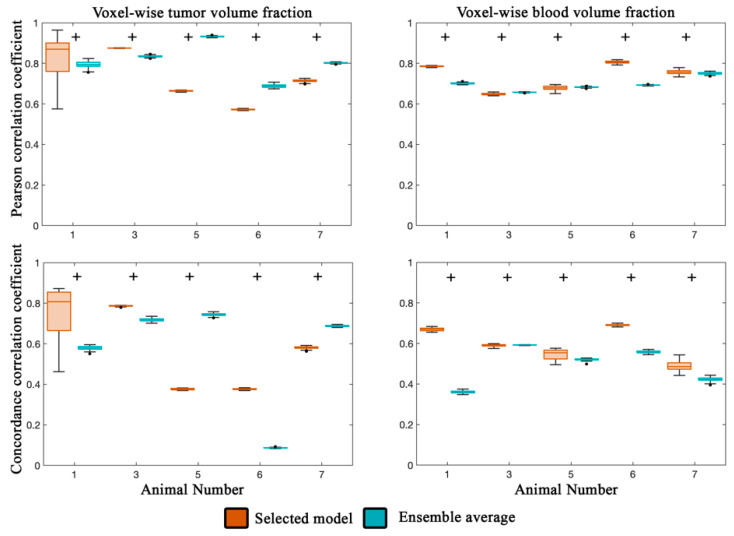
Local error analysis from the prediction scenario. The Pearson correlation coefficient (top row) and concordance correlation coefficient (bottom row) are reported for the tumor (left column) and blood volume (right column) fractions for the prediction scenario for five animals. (Animals 2 and 4 had insufficient imaging visits for the prediction scenario). Lower PCCs were observed for animal 5 and 6 for the selected model compared to the rest of the cohort. Elevated PCCs were observed for three out of five animals. Similarly, voxel level agreement was greater than 0.58 for animals 1, 3, and 7 for the selected model. Blood volume predictions at the voxel level had a high level of correlation (PCCs greater than 0.65) for both the selected and ensemble averaged model. Lower agreement was observed between the predicted and measured blood volume resulting in CCCs greater than 0.49 for the selected model and 0.36 for the ensemble model.

**Figure 6 cancers-13-01765-f006:**
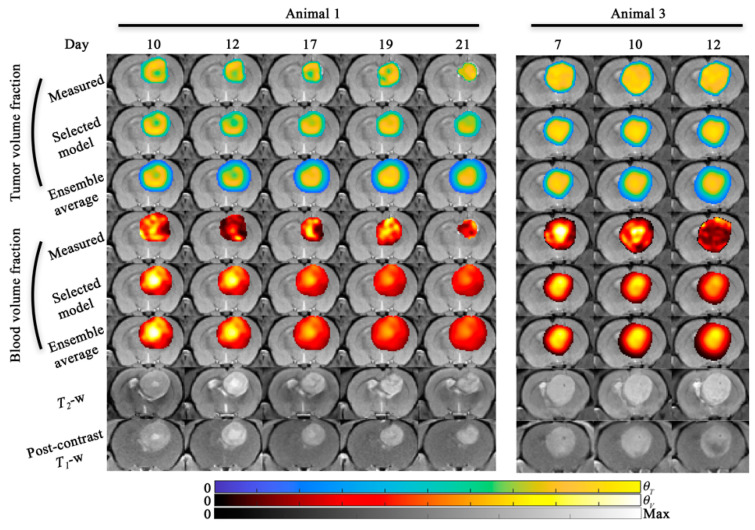
Model predictions (scenario 2) for animal 1 and 3. Model predictions of tumor and blood volume fractions at the central slice are shown for animal 1 and animal 3. For both animals, the model predictions from the selected model and the ensemble average are shown. The bottom two rows show the *T*_2_-weighted and post-contrast *T*_1_-weighted MRI. For animal 1, both the selected and ensemble average model predict the areas of intratumor heterogeneity in the tumor volume predictions. Blood volume predictions tended to predict increased blood volume in the interior of the tumor relative to the periphery as observed. For animal 3, the observed tumor is more homogeneous (compared to animal 1) and the model is able to predict this distribution. Additional, both models predict a higher blood volume fraction towards the interior of the tumor at the final time point which is not present in the measured blood volume fraction. For animal 1, tumor growth was predicted from day 10 to 21 and resulted in −1.0% and 261% error in tumor volume predictions for the selected and ensemble models, respectively. For animal 3, both the selected and ensemble average model resulted in <4.6% error in tumor volume predictions. A sagittal view of this figure is shown in Appendix A.

**Figure 7 cancers-13-01765-f007:**
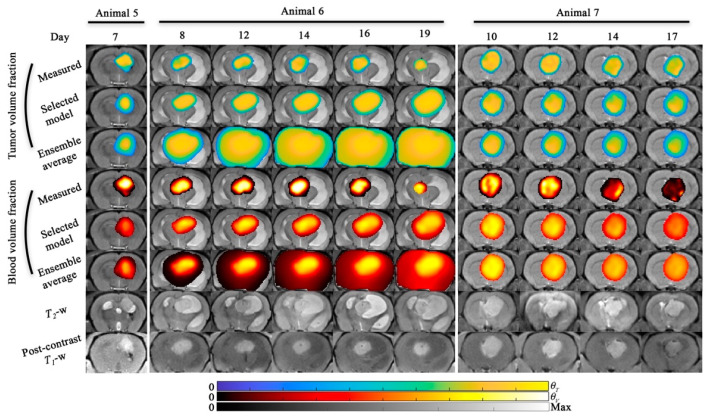
Model predictions (scenario 2) for animal 5, 6, and 7. Model predictions of tumor and blood volume fractions at the central slice are shown for animal 5, 6, and 7. For all animals, the model predictions from the selected model and the ensemble average are shown. The bottom two rows show the *T*_2_-weighted and post-contrast *T*_1_-weighted MRI. For animal 5, only one prediction time point was available and both the selected and ensemble average model resulted in <8.7% error in tumor volume predictions. For animal 6, tumor growth was predicted from day 8 to 19 and resulted in 16.0% and 521% error in tumor volume predictions for the selected and ensemble models, respectively. For animal 7, four prediction time points were available and both the selected and ensemble average model resulted in <21.5% in tumor volume predictions across all time points. A sagittal view of this figure is shown in Appendix A.

**Figure 8 cancers-13-01765-f008:**
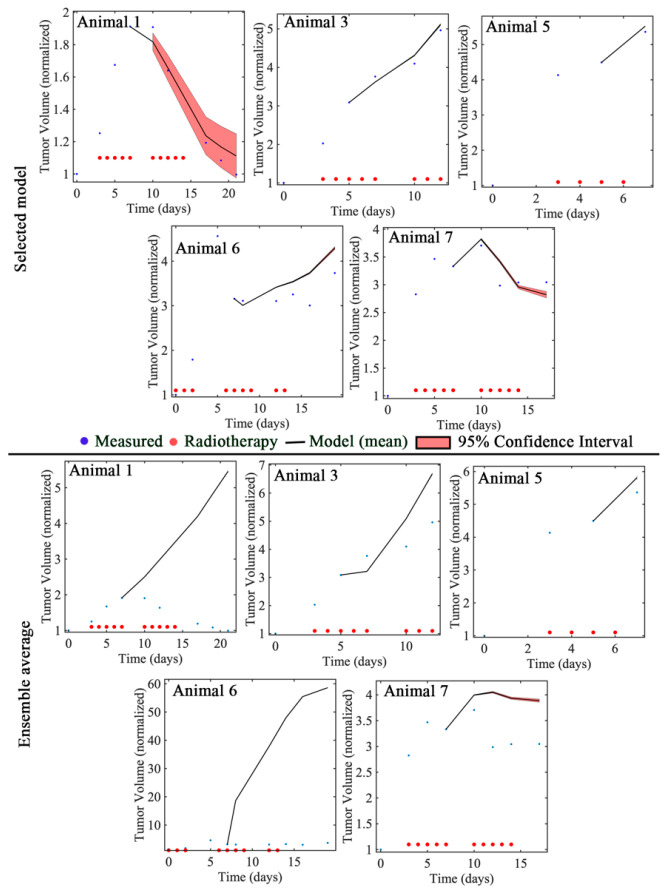
Tumor volume prediction time courses (scenario 2). For each animal the measured (blue dots), model mean (black line), model 95% confidence interval (shaded red), and dates of radiotherapy (red dots) are shown for both the selected model (top panel) and ensemble average (bottom panel). For the selected model, larger confidence intervals were observed for animal 1 relative to the other models. For the ensemble average, tumor growth was generally overestimated.

**Table 1 cancers-13-01765-t001:** Model parameters and variables.

Parameter or Variable Group	Parameter or Variable	Interpretation	Source
Measured	$\phi_{T}\left(\bar{x},t\right)$	Tumor cell volume fraction	DW-MRI
	$\phi_{V}\left(\bar{x},t\right)$	Blood volume fraction	DCE-MRI
	*θ_min_*	Minimum value for carrying capacity	DW-MRI
	*θ_V_*	Maximum blood volume	DCE-MRI
	$\bar{\phi_{V,pre-treatment}}$	$\mathrm{Average}\;\mathrm{pre}-\mathrm{treatment}\;\phi_{V}\left(\bar{x},t\right)$	DCE-MRI
Calibrated	*k* _*p,T*,0_	Tumor cell proliferation rate	Calibrated globally or locally
	*kd*,_*T*,0_	Tumor cell death rate	Calibrated globally
	*D*_*T*,0_, *D*_*V*,0_	*ϕ_T_* and *ϕ_V_* diffusion coefficients in the absence of mechanically coupling	Calibrated globally
	*ϕ* _*V,thresh*_	Threshold on *ϕ_V_* for carrying capacity to decrease	Calibrated globally
	*θ_max_*	Max carrying capacity	Calibrated globally
	*k_p,V_*	Vasculature proliferation rate	Calibrated globally
	*k_d,V_*	Vasculature death rate	Calibrated globally
	*SF*	Surviving fraction	Calibrated globally
	*α* _1_	Coupling constant	Calibrated globally
Assigned	*d*	Distance to the periphery of the tumor	Calculated
	$\theta_{T,V}\left(\bar{x},t\right)$	Combined *ϕ_T_* and *ϕ_V_* carrying capacity	Calculated
	*λ* _1_	Coupling Constant	Set to 0.25
	*λ* _2_	Coupling Constant	Set to 1
	*G, ν*	Shear modulus and Poisson’s ratio assigned for white and gray matter	Literature [30]

**Table 2 cancers-13-01765-t002:** Experimental timeline.

Animal Number	Dose/Day	Image Days	Treatment Dates	Calibration Dates	Prediction Dates
81	2 Gy	0,3,5,7,10, 12,17,19,21	3–7,10–14	0–7	10–21
2	4 Gy	0,3,5	3–5	0–5	NA
3	2 Gy	0,3,5,7,10,12	3–7,10–12	0–5	7–12
4	2 Gy	0,3,5	3–6	0–5	NA
5	2 Gy	0,3,5,7	3–6	0–5	7
6	4 Gy	0,2,5,7,8, 12,14,16,19	0–2,6–9,12,13	0–7	8–19
7	4 Gy	0,3,5,7,10, 12,14,17	3–7,10–14	0–7	10–17

**Table 3 cancers-13-01765-t003:** Calibrated model parameters for selected model.

Parameter or Variable	Animal Number (Dose per Day)						
	1 (2 Gy)	2 (4 Gy)	3 (2 Gy)	4 (2 Gy)	5 (2 Gy)	6 (4 Gy)	7 (4 Gy)
*k_p,T,_*_0_ (day^−1^)	0.84 ± 0.25	1.20 ± 0.35	1.40 ± 0.48	1.10 ± 0.16	1.82 ± 0.21	4.65 ± 5.45	2.04 ± 0.43
*kd_,T,_*_0_ (day^−1^)	0.22 ± 0.04	0.40 ± 0.07	0.11± 0.01	0.03 ± 0.06	0.5 ± 0.10	0.35 ± 0.04	NA
*D_T,_*_0_ (10^4^ m^2^ day^−1^)	2.32 ± 0.07	4.50 ± 0.07	2.58 ± 0.07	1.37 ± 0.02	3.44 ± 0.05	6.93 ± 0.30	3.38 ± 0.02
*D_V,_*_0_ (10^4^ m^2^ day^−1^)	2.72 ± 0.06	3.00 ± 0.09	3.10 ± 0.09	3.00 ± 0.19	3.11 ± 0.05	3.25 ± 0.12	3.00 ± 0.04
*ϕ_V,thresh_*	0.03 ± 0.12	0.05 ± 0.06	0.03 ± 0.05	0.03 ± 0.02	0.02 ± 0.04	0.02 ± 0.02	0.02 ± 0.08
*θ_max_*	0.90 ± 0.01	0.90 ± 0.02	0.94 ± 0.02	0.87 ± 0.03	0.92 ± 0.01	0.91 ± 0.00	0.92 ± 0.01
*k_p,V_* (day^−1^)	0.65 ± 0.03	1.10 ± 0.06	0.82 ± 0.04	1.25 ± 0.17	1.49 ± 0.13	0.17 ± 0.47	1.76 ± 0.02
*k_d,V_* (day^−1^)	0.18 ± 0.02	0.32 ± 0.06	0.29 ± 0.04	0.54 ± 0.10	0.97 ± 0.17	0.07 ± 0.04	0.07 ± 0.04
*SF*	0.95 ± 0.01	0.80 ± 0.08	0.97 ± 0.01	0.99 ± 0.01	0.98 ± 0.02	0.90 ± 0.01	0.91 ± 0.01
*α* _1_	NA	NA	4.00 ± 0.17	NA	NA	NA	NA

The mean 95% confidence interval are reported for each parameter. NA indicated that the parameter was not estimated for the selected model.

## Data Availability

The datasets generated or analyzed, and the code used during the current study are available from the corresponding author on reasonable request.

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
