# Peer review of "Towards an Image-Informed Mathematical Model of In Vivo Response to Fractionated Radiation Therapy"

_cancers, 2021, doi:10.3390/cancers13081765_

Round 1
Reviewer 1 Report
The authors present a well-written manuscript describing a modelling approach for the prediction of in vivo tumour growth following fractionated radiotherapy informed by quantitative magnetic resonance imaging. The modelling approach presented is very thorough, and clinically highly relevant making this study well suitable for publication. I have highlighted fairly minor corrections with the exception of the evaluation of the delivered vs. planned radiation dose, but a am hopeful that the authors can easily address these comments. This is a very interesting study providing high quality evaluation of a mathematical modelling approach providing spatio-temporal estimations of tumour response. This could indeed provide a very promising approach on the way towards personalized RT. As such a suggest that the study would be highly relevant to the readers of Cancers.
The authors present a well-written manuscript describing a modelling approach for the prediction of in vivo tumour growth following fractionated radiotherapy informed by quantitative magnetic resonance imaging. The modelling approach presented is very thorough, and clinically highly relevant making this study well suitable for publication after addressing the below comments. In general, some of the used (mostly experimental) methods require some clarification and I would further recommend to provide some additional aspects in terms of results and discussion.
Methods:
- Indicate location of IC injection and where in the brain tumours were observed. How were injections performed? Cite references if this method has previously been described by your lab. Give more details here, including authentication of cell lines (STR analysis) and mycoplasma status, as well as culture conditions prior to injection in brief.
- Indicate how dose at the tumour location was calculated to ensure planned and delivered doses agreed. If available provide an estimated dose distribution and animal imaging in treatment position with indicated beam angles. At which angles was treatment delivered? Give details on RT delivery calibration and potential beam filtration (can be part of supplement).
- Indicate the software/platform used to perform image registration and how a registration was performed/scored a success.
- Who performed image segmentation (medical doctor?), and what was included as part of the segmented tumour? Provide an estimate of contouring uncertainty.
- It appears that a uniform surviving fraction was assumed. This may be counter intuitive in case the delivered dose distribution was not perfectly uniform which could be the case for a single angle delivery. Please indicate in Table 1 which parameters were assumed to be globally constant, or spatially variable, as well as the parameter range used for model fitting.
Results:
- Show longitudinal tumour volume for both measurement and model predictions/fit in addition to percent errors and spatial distributions. How did errors evolve over time? It would be expected that earlier time points would be predicted better than later ones, unless there is no differences showing averages only may be misleading.
- Provide a sagittal view of the results shown in figures 6,7, too (potentially supplementary material).
- Provide the obtained calibrated model parameters with 95% confidence intervals, compare them between subjects (and dose regimes) and, where applicable, with relevant literature data in the discussion.
- It would strengthen the potential application of the model if a simulation of an alternative spatial dose distribution and/or temporal scheduling would be provided.
Discussion:
- Please discuss potential reasons for the far poorer prediction of animal 6 compared to the other subjects and potential mitigation strategies, as well as indications for potential patient treatment.
- The selected model seems to provide a good short-term prediction of tumour growth in most cases, however long-term treatment response evaluation, e.g. prediction of time to progression, may be clinically more relevant. It is clear that such analysis may not be possible with the relevant data, however this may be included in the outlook, and also motivate the delivery of more than 10 treatment fractions. Discuss this point also with respect to the change of model performance over time.
- It seems surprising that there was no difference in response in the 2Gy and 4Gy treated subjects. Please comment on this and provide a potential explanation.
Minor comments:
- p. 3, l. 89 and l.107: The LQ model itself does not include any dynamic information, and hence does not assume instantaneous cell death per se. It is derived based on clonogenic assay data that are evaluated several days post RT and hence no dynamic evaluation is included (as the authors state themselves in the first part of the introduction). As such, it does not include the assumption of instantaneous cell kill. Please rephrase the sentence ‘…that the LQ model may be more appropriate for long time scales’ – e.g. ‘…that modelling instantaneous cell kill using a fraction of surviving cells calculated by the LQ model…” and also (l. 107) “similar to the standard LQ approach”.
- Figure 6, 7: include scale bar, and indicate imagine contrast displayed as well as an indication of the relevant (coronal) slice with respect to the animal. Add a row showing the structural, contrast enhanced MRI.
- p. 16 l. 522 repetition of word ‘evaluate’
- Caption Figure 6 l.502 – animal 1 rather than 6?
- Mention the limitation of using immunocompromised animals.
- The code used for the analysis and modeling should be made publicly available.
Author Response
Please see the attachement.

Reviewer 2 Report
Please find my report in the attached file.

Round 2
Reviewer 2 Report
The authors have addressed the comments and recommendations made in my first report, and included the pertinent changes in their revised version, which has been much improved. Their reply to the main criticism raised in my first report, although not entirely satisfactory, has been reasonably argued and it is indicative that their work will be further pursued in the future along the lines suggested.
I therefore do recommend publication of the revised manuscript.
Author Response
We thank the reviewer for their critiques, suggestions, and willingness to review this manuscript.